# Application of Acidulants to Control *Salmonella* spp. in Rendered Animal Fats and Oils with Different Levels of Unsaturation

**DOI:** 10.3390/ani13081304

**Published:** 2023-04-11

**Authors:** Janak Dhakal, Charles G. Aldrich

**Affiliations:** 1Department of Agriculture, Food and Resource Sciences, University of Maryland Eastern Shore, Princess Anne, MD 21853, USA; jdhakal@umes.edu; 2Department of Grain Science and Industry, Kansas State University, Manhattan, KS 66506, USA

**Keywords:** *Salmonella* spp., sodium bisulfate, organic acids, chicken fat, canola oil, Menhaden fish oil, lard, tallow

## Abstract

**Simple Summary:**

Despite the US Food and Drug Administration’s zero-tolerance policy against *Salmonella*, several *Salmonella*-linked outbreaks and recalls linked to pet foods have been reported. Post-processing steps, such as fat and flavor coating, drying, cooling, and packaging are common steps where *Salmonella* becomes contaminated in dry pet food kibbles. Rendered animal fats and oils are commonly coated on kibbles to enhance palatability and increase energy density. A tiny layer of water in the bottom of bulk fat transport trucks or/and storage tanks could easily harbor *Salmonella*, leading to its entry into the pet food during coating. In this study, different types of acidulants were applied in the fat and oil system, and their effect against *Salmonella* in the water or fat phase in different types of fats and oils was evaluated. Our results were promising, wherein all the tested acidulants were effective to mitigate *Salmonella* from the fat or oil system (both aqueous and fat phase) within 2 h. The findings of this study could be helpful to the rendering and pet food industry in a fight to mitigate *Salmonella* in pet food.

**Abstract:**

*Salmonella*-contaminated pet foods could potentially become a source of human salmonellosis. This study evaluated the survival of *Salmonella* without and with the addition of acidulants in different fat types (chicken fat (CF), canola oil (CO), Menhaden fish oil (FO), lard (La), and tallow (Ta)) commonly used to coat dry pet food kibbles. The minimum inhibitory concentration (MIC) of individual acidulants and the combination were determined using the broth microdilution method. Autoclave-sterilized rendered fats were treated with pre-determined concentrations of antimicrobial acidulants (0.5% sodium bisulfate (SBS), 0.5% phosphoric acid (PA), 0.25% lactic acid (LA), etc.) and incubated overnight at 45 °C. The treated fats were inoculated with approximately eight logs of a *Salmonella* cocktail. Microbiological analyses were conducted separately for the fat-phase and water-phase at predetermined time intervals (0, 2, 6, 12, and 24 h) by plating them onto TSA plates. After incubating at 37 °C for 24 h, the plate count results were expressed as log CFU/mL. The MIC of SBS was 0.3125%, and of PA and LA were both 0.1953% against cocktail *Salmonella* serotypes. We observed a possible synergistic effect when SBS and organic acid were combined. All the acidulant tested at targeted concentrations individually as well as in combination with organic acids were highly effective against *Salmonella* spp. (non-detectable within 2 h) across different fat types. A potent anti-bactericidal effect leading to non-detectable *Salmonella* immediately (<1 h) at 45 °C was observed in the aqueous phase of the fish oil system, even without the addition of acidulants. These findings are significant for the dry pet food industries, where potential post-processing contamination of *Salmonella* could be controlled by treating fats and oils with acidulants.

## 1. Introduction

The use of rendered animal fats as a value-added material in pet food is a common practice. During the rendering process, heat is applied, moisture is removed, and fats are separated. However, food safety concerns arise as animals used in the rendering process are natural microbiological reservoirs, including human pathogens such as *Salmonella* spp. *Clostridium perfringens, Listeria monocytogenes*, and *Campylobacter jejuni* [1]. 

Continued improvements within the industry have implemented process control monitoring to ensure proven cook times and temperatures have been reached for the inactivation of specific microorganisms deemed to be a food safety hazard [2]. While the rendering industries have an aggressive approach to animal food ingredient quality and safety by using long cook times and high temperatures, contamination with pathogenic microorganisms still occurs. Studies have shown the presence of *Salmonella* spp. in final rendered products, including protein meals, meat, and bone meal, feather meal, meat meal, and poultry meals [1,3,4,5,6,7]. However, the evaluation of microbial contamination of rendered fats, specifically poultry fat, beef tallow, or other animal fat products, was not included in these surveys. While the rendering process itself is effective in killing pathogens as observed with 0% *Salmonella* spp. in crax [8], it is a point-in-time mitigation strategy bearing no residual activity. Post-processing contamination from dusts, equipment and humans is considered to be the main source of *Salmonella* spp. introduction in final rendered products [1,5,8]. Contaminated rendered products have the potential to contaminate animal feed and pet foods. For example, due to the high *Salmonella* load on chicken offal, viscera, and animal co-products, the incoming raw materials in chicken fat rendering facilities might become a source of *Salmonella* contamination in the facility and premises if the cleaning and sanitation are not proper. In fact, several outbreaks of human *Salmonella* infection have been traced back to contaminated animal feed [9,10,11,12]. Application of proper processing temperature and holding time and the use of quality raw ingredients are prime to reduce the risk of pathogen contamination in feed production. Promising research by Cochrane et al. [13] has shown effective mitigation of *Salmonella* through the addition of chemical additives to rendered animal proteins, including feather meal, avian blood meal, porcine meat, bone meal, and poultry by-product meal.

Dry pet food constitutes the most sold pet food type in the US; USD 5.99 billion worth of dry dog food and USD 2.84 billion worth of dry cat food were sold in 2021 in the USA [14]. Rendered animal fats are commonly incorporated into dry pet food kibbles for added energy, essential fatty acids, and as a palatant. Higher fat content and low water activity in rendered animal fats, both exhibiting bacteriostatic properties, are parameters used in foods to curb microbial growth. However, two major multistate outbreaks of human *Salmonella* Schwarzengrund and *S.* Infantis in 2008 and 2012, respectively, were sourced back to contaminated dry pet foods. The pathogens were assumed to have entered during the flavoring and enrobing steps in the coating process. The bulk fats and oils used in pet food industries are transported and stored in bulk trucks and tanks. It is common that these commercial fats hold a small (~3%) moisture, insoluble, and unsaponifiables (MIU).

Acidulants, including sodium bisulfate, phosphoric acids, and organic acids, are known to have an antimicrobial effect against various bacterial pathogens, including *Salmonella*. Our previous studies have shown that sodium bisulfate (SBS), lactic acid, and phosphoric acid were potent against *Salmonella* in rendered chicken fat [15]. The fats and oils used to coat dry pet food kibbles differ in the composition of fatty acids. The effect of the composition of fats and oils to mitigate *Salmonella* has not been evaluated. Therefore, this study aimed to evaluate the survival of outbreak-linked *Salmonella* serotypes in fats and oils (chicken fat, canola oil, Menhaden fish oil, lard, and tallow) differing in saturation level and treated with food-grade acidulant antimicrobials, such as sodium bisulfate (SBS), lactic acid (LA), phosphoric acid (PA), and a combination of SBS with organic acids (butyric acid (BA), lactic acid (LA), and propionic acid (PrA).

## 2. Materials and Methods

### 2.1. Salmonella Serotypes, and Fats/Oil Sources

*Salmonella* Enteritidis (ATCC 4931; source, gastroenteritis), *Salmonella* Heidelberg (ATCC 8326), and *Salmonella* Typhimurium (ATCC 14028; source, poultry) were used in the experiment and were maintained in tryptic soy broth (TSB)-glycerol (7:3) at −80 °C. Prior to use, the frozen cultures were streaked on tryptic soy agar (TSA) plates and incubated at 37 °C for 24 h. A single colony of *Salmonella* strain was inoculated in 10 mL of TSB and incubated at 37 °C for 18 to 24 h. An equal volume of each serotype, incubated for the same duration, was mixed to make a cocktail of three serotypes. 

Chicken fat and beef tallow were provided by Darling Ingredients (Irving, TX, USA). Kroger^®^ Pure Canola Oil was purchased from a local Kroger store. Rendered Menhaden fish oil was provided by an established fish rendering company (Omega Protein^®^, Inc., Reedville, VA, USA). Morrell snow cap lard was provided by John Morrell & Co. (Cincinnati, OH, USA).

### 2.2. Minimum Inhibitory Concentration

The minimum inhibitory concentration (MIC) of individual acidulants and combinations was determined using the broth microdilution method [16]. A single colony of each of the three *Salmonella* serotypes (*S*. Enteritidis (ATCC 4931), *S*. Heidelberg (ATCC 8326), and *S*. Typhimurium (ATCC 14028)) and the cocktail were inoculated in 10 mL of Tryptic Soy Broth (TSB) and incubated at 37 °C for 18 to 24 h. The overnight-grown culture was centrifuged at 5000 rpm for 10 min at room temperature, and the bacterial pellet was re-suspended in fresh TSB.

A volume of 200 µL of an antimicrobial solution consisting of twice the desired final concentration was dispensed in the first well of a 96-well plate (triplicate wells), and 100 µL of sterile water in the rest of the wells. Serial two-fold dilutions of the antimicrobial solutions were performed. One hundred microliters of bacterial culture containing ~5 logs CFU/mL (in 2X TSB) were added to each well to make a final volume of 200 µL. A positive control consisted of *Salmonella* inoculum only (no antimicrobial), and a negative control consisted of TSB without *Salmonella* and an antimicrobial agent. The microtiter plate was incubated at 37 °C for 24 h. The MIC was determined to be the lowest concentration of an antimicrobial that inhibited the visible growth of *Salmonella* after 24 h of incubation at 37 °C.

### 2.3. Survival of Salmonella in Fats and Oils

For this study, a cocktail of three serotypes was tested in the fat and oil system, both with and without the presence of acidulant antimicrobials. The fat or oil system refers to a mixture of fat or oil with water consisting of a bottom water phase and an upper fat phase for any specific fat/oil type. Based on the MIC of the acidulants and their combination, six different acidulants and their combinations were used to treat the oil and fat system (Table 1). Autoclave-sterilized rendered fats and oils were treated with pre-determined concentrations of antimicrobial acidulants and left overnight at 45 °C. Because we were analyzing the two phases separately, we needed the fat or oil system in molten state, and incubation at 45 °C effectively kept all the fat types in molten state. The acidulants were treated in aqueous form, and ~3% moisture was maintained in all the treatments, thereby creating an aqueous and a fat phase in the fat and oil system. The antimicrobial treated fats were then inoculated with ~8 logs of a *Salmonella* cocktail. The volume of the acidulants and bacterial suspension were adjusted in a way to achieve ~3% final moisture percentage in the fat system. Microbiological analyses were conducted separately for the fat-phase and water-phase treatments at predetermined time intervals (0, 2, 6, 12, and 24 h). From each subsample, the fat phase and aqueous phases were gently removed by pipetting, diluted in 0.1% peptone water (pre-warmed at 45 °C), and plated onto TSA nutrient plates. For the fat phase, samples were diluted and dispensed onto the agar plates by vigorously vortexing the tubes and removing 100 μL while the suspension was still in the emulsion. A negative control containing fats/oils alone, without antimicrobial or *Salmonella* spp., was maintained. Fats treated with sterile distilled water instead of acidulant antimicrobials were maintained as controls for each fat type. The plates were incubated at 37 °C for 24 h, and then colonies were counted with results expressed as log CFU/mL. The limit of detection is 1 log CFU/mL in this study. The experiment was a 6 × 5 factorial arrangement of treatments utilizing five fat types and five sampling intervals. The experiment was evaluated in triplicate.

## 3. Results

### 3.1. Minimum Inhibitory Concentration

The minimum inhibitory concentration (MIC) of acidulant antimicrobials, both individually as well as in combination, against three serotypes of *Salmonella*, both individually as a cocktail of three serotypes, were determined and summarized in Table 2. Higher minimum concentrations of SBS, LA, and PA were required to inhibit *Salmonella* Typhimurium, as compared to *S*. Heidelberg and *S*. Enteritidis: (0.50% vs. 0.31%), (0.50% vs. 0.20%), and (0.25% vs. 0.10%), respectively. Similarly, for the cocktail of three serotypes, LA and PA were more effective (MIC 0.20%) compared to SBS (MIC 0.31%). When a combination of SBS with lactic acid was used, we observed a potential synergistic effect against individual as well as cocktail serotypes.

### 3.2. Survival of Salmonella in Fats and Oils

The effects of 0.5% SBS treatment on *Salmonella*-inoculated fats and oils over time are represented in Figure 1. The fat or aqueous phases showing non-detectable *Salmonella* count at 0 h are not shown in figures (Figure 1, Figure 2, Figure 3, Figure 4, Figure 5 and Figure 6). In the fat phase of the SBS-treated chicken fat system, the *Salmonella* count was reduced to 0.77 logs within 0 h and was reduced to a non-detectable level by 2 h, whereas in the aqueous phase, *Salmonella* was reduced to a detectable level at 0 h. In the case of the control (without SBS) chicken fat system, the *Salmonella* count was reduced to 2.60 logs and remained at 7.60 logs at 0 h in the fat phase and aqueous phase, respectively. After 2 h, *Salmonella* was non-detectable in the fat phase but remained >7.13 logs until 24 h in the aqueous phase. The addition of SBS in canola oil also led to *Salmonella* reduction to a non-detectable level in the aqueous phase and to 1.53 logs in the fat phase within 0 h. In the untreated control canola oil system, the *Salmonella* count in the fat phase reduced to 0.34 log at 0 h and to a non-detectable level at 2 h, whereas in the aqueous phase, it remained >7.90 logs throughout the incubation. In Menhaden fish oil, in both SBS-treated and control fat systems, *Salmonella* was reduced to a non-detectable level by 0 h in both fat phases and water phases. Furthermore, SBS-treated lard reduced *Salmonella* to a non-detectable level in both the fat phase and aqueous phase at 0 h. Similarly, the *Salmonella* count was non-detectable at 0 h in the fat phase of the control lard system but remained >7.72 logs throughout the 24 h study period. Beef tallow, when treated with SBS, led to *Salmonella* reduction to a non-detectable level within 0 h in both fat and aqueous phase. Whereas in the control tallow system, though *Salmonella* lowered to a non-detectable level at 0 h in the fat phase, it remained >7.63 logs throughout the 24 h study period in the aqueous phase. 

The effect of 0.5% phosphoric acid (PA) treatment on *Salmonella*-inoculated fats and oils over time is represented in Figure 2. The aqueous phase of all the control fat and oil systems, except Menhaden fish oil, was able to contain a *Salmonella* level between 7.89 logs (in tallow) to 8.27 log CFU/mL (in chicken fat) through the 24 h incubation period. Whereas 1.59 logs of *Salmonella* were recorded in untreated control Menhaden oil at 0 h, which reduced to a non-detectable level by 2 h. The *Salmonella* dropped to a non-detectable level in PA-treated Menhaden fish oil system (both aqueous and fat phase) at 0 h. 

The effect of 0.25% lactic acid treatment on *Salmonella*-inoculated fat and oils over time is represented in Figure 3. In the control fat and oil system, *Salmonella* remained between 7.57 logs (in lard) to 8.36 logs (in canola oil) throughout the incubation period in the aqueous phase, except for Menhaden fish oil, where the *Salmonella* count dropped to 0.59 log at 0 h and went to a non-detectable level by 2 h. In the case of untreated (control) fat phase, the *Salmonella* was non-detectable at 0 h in lard and Menhaden fish oil. Whereas *Salmonella* counts were 4.37, 2.36, and 0.33 logs in chicken fat, canola oil, and tallow, respectively, which declined to a non-detectable level by 2 h. In the lactic acid-treated fat and oil system, *Salmonella* was non-detectable in the aqueous phase of all fat systems within 0 h. By 2 h, the *Salmonella* was non-detectable in fat phases of all fats and oil systems.

The effect of a combination of 0.1% SBS + 0.075% BA treatment on *Salmonella*-inoculated fats and oils over time is shown in Figure 4. In the aqueous phase of control fat and oil systems, *Salmonella* remained between 7.74 logs (in canola oil) to 8.17 logs (in tallow) throughout the incubation period, except for Menhaden fish oil where the *Salmonella* count reduced to a non-detectable level within 0 h. *Salmonella* levels in untreated (control) fat phases in all fats and oils reduced to non-detectable by 2 h. In the case of 0.1% SBS + 0.075% BA treated fat and oil system, except for chicken fats, *Salmonella* levels were dropped to a non-detectable within 0 h. In chicken fat, both in fat and aqueous phases, *Salmonella* was non-detectable in 2 h. 

Figure 5 shows the effect of a combination of 0.15% SBS + 0.1% LA treatment on *Salmonella*-inoculated fat and oils over time. In the aqueous phase of the control fat and oil systems, *Salmonella* levels remained between 7.57 log (in chicken fat) and 8.10 log (in canola oil and Lard) throughout the incubation period, except for the Menhaden fish oil where the *Salmonella* count was reduced to a non-detectable level within 0 h. In the fat phases of all the untreated (control) oil and fat systems, the *Salmonella* level was reduced to a non-detectable level in 2 h. In 0.15% SBS + 0.1% LA treated fats and oil systems, a reduction pattern similar to 0.1% SBS + 0.075% BA was observed. 

Similarly, the effect of a combination of 0.1% SBS + 0.05% PrA treatment on *Salmonella*-inoculated fat and oils over time is represented in Figure 6. The *Salmonella* count was recorded between 7.99 logs (in chicken fat) to 8.19 logs (in tallow) during the 24 h incubation period in the aqueous phase of the control fat and oil systems, except for Menhaden fish oil where the *Salmonella* level dropped a non-detectable level at 0 h. 

*Salmonella* was non-detectable by 2 h in both the phases of the acidulants-treated fat and oil systems (Table 1) and the fat phase of control fat-oil systems (Table 3). All the treatments, namely 0.5% SBS, 0.5% PA, 0.25% LA, and a combination of SBS with organic acids, were effective in lowering the *Salmonella* load in both the aqueous phase and fat phase of the fat system to a non-detectable level within 2 h.

## 4. Discussion

Rendered animal fats and oils vary in the amount of unsaturated fatty acids they contain, and these unsaturated fatty acids are known to exhibit antimicrobial effects [17]. Unsaturated fatty acids are known to have greater antimicrobial activity against *Vibrio* spp., a gram-negative bacterium than saturated fatty acids [18]. However, saturated fatty acids such as caproic, caprylic, and capric acids also exhibited potent antimicrobial activity against *Salmonella* [19]. Similarly, it was reported that unsaturated fatty acids such as linolenic and myristoleic acids exerted potent antibacterial activity against *H. pylori* [20], and linolenic acid against *Bacillus cereus* and *Staphylococcus aureus* [21].

In this study, *Salmonella* was non-detectable by 2 h in the fat phase of both acidulant-treated as well as non-treated (control) fat systems. The reduction in *Salmonella* from the acidulant-treated fat and oil system could be explained by the antimicrobial effects of acidulants. However, similar findings from the control fat and oil systems indicated the role of the innate antimicrobial effect of fats and oils, especially unsaturated fatty acids content. In addition, the possibility of the presence of *Salmonella* in a non-detectable form, for example, a viable but non-culturable state (VBNC), could not be negated [22]. However, the reduction in *Salmonella* from the aqueous phase of the antimicrobial-treated fat system could be credited to the potent action of acidulant antimicrobials which were applied as an aqueous inoculum. A Prior study from our lab validated the effectiveness of acidulants such as SBS in mitigating *Salmonella* from rendered chicken fat [15]. 

In the case of canola oil and beef tallow, individual treatment of SBS, phosphoric acid, and lactic acids took 2 h to lower *Salmonella* to a non-detectable limit (with the exception of 0.5% SBS treatment in tallow), whereas a combination of SBS and organic acids caused an immediate (0 h) reduction in *Salmonella* to a non-detectable level (Table 2). The latter could potentially be due to the synergistic action of SBS and organic acid. In addition, high levels of polyunsaturated fatty acids (PUFA) in canola oil (31.3%) and tallow (20.3) might also have contributed to the enhanced antimicrobial effect. Similar synergistic effects of SBS and organic acids were also observed against *Salmonella* in the chicken fat system [15]. The synergistic effect could be supported by the fact that both SBS and organic acids act against bacteria by damaging the cell membrane and causing oxidative damage to the cells [23,24].

Molitor et al. [25] reported that chicken fat contains 36.8% oleic acid (18:1), 21.1% linoleic acid (18:2), and <1% each of alpha-linoleic acid, (18:3:3), eicosenoic acid (20:1), eicosadienoic (20:2), and arachidonic acid (20:4). The level of unsaturated fatty acids in CO was highest, with only less than 8% being saturated fatty acids. However, the majority (~65%) of the unsaturated fatty acids are monounsaturated (oleic acid), and 26.3% are polyunsaturated fatty acids (PUFA), with only 8.6% are linolenic acid, and 17.7% being Linoleic acids [26]. In a study by Deliephan et al. [27], when a mixture of 2-Hydroxy-4-(Methylthio) Butanoic Acid (HMTBa), lactic acid, and phosphoric acid incorporated into canola oil were coated on dry dog food kibbles, *Salmonella* spp. (Enteritidis, Heidelberg, and Typhimurium) were reduced to a non-detectable level from their initial concentration of approximately eight logs within 72 h. In a related study, the authors also reported that the organic acid mixtures containing HMTBa effectively mitigate *Salmonella* from food contact surfaces (plastic, rubber, stainless steel, and concrete) commonly used in pet food industry [28]. Phosphoric acid (PA) seemed to be less effective (non-detectable in 2 h) in the aqueous phase as compared to the other acidulants (non-detectable in 0 h) in the canola oil fat system. This is contrary to the effectiveness of phosphoric acid we observed in vitro MIC assay, where a lower concentration of PA was needed to inhibit the visible growth of *Salmonella* as compared to the SBS and LA. This could partly be explained by the reduced potency of inorganic acid, i.e., phosphoric acid, in the complex organic matrix of the canola oil system. 

Menhaden fish oil contains a high amount of unsaturated fatty acids, with 34.8% being PUFA (~29.41% omega 3), and 23.46% monounsaturated fatty acids [29]. Alpha-linoleic acid is the most common omega-3. The reduction in *Salmonella* to a non-detectable level within 0 h in both fat and aqueous phases of treated FO samples (Table 2) could be explained by the combined effect of the antimicrobial acidulants and the presence of high PUFA [30]. We also observed a very interesting result where *Salmonella* in the aqueous phase of untreated (control) Menhaden fish oil system was mitigated to a non-detectable level immediately. Out of the six different sets of experiments with different acidulants, *Salmonella* level in the aqueous phase of non-acidulant-treated Menhaden fish oil dropped to a non-detectable level immediately (0 h) in five of them. Whereas, in one (0.25% lactic acid) treatment, 0.59 log of *Salmonella* was detected at 0 h, which was reduced to a non-detectable level at 2 h. This minor deviation in only one set could be attributed to sampling or handling variations. Similar results with fish oil were reported in a past study by our lab [22]. As we hypothesized, the absence of detectable *Salmonella* from the aqueous phase of control Menhaden fish oil samples indicated that the higher the unsaturation level, the more antimicrobial activity an oil/fat possesses, which was similar to the findings by Knapp and Melly [31].

Beef tallow is considered a saturated fat and contains 35–64% unsaturated fatty acids [32,33], with 38.6% oleic acid (18:1), 20.3% linoleic acid (18:2), and <1% each of alpha-linoleic acid, (18:3:3), eicosenoic acid (20:1), eicosadienoic (20:2), arachidonic acid (20:4) [25]. Native Malaysian pork lard contains 45–62% unsaturated fatty acids [32,33], with 17.3% linoleic acid [33]. The high amount of unsaturated fatty acids in lard could have contributed to the reduction in *Salmonella* to a non-detectable level within 2 h in the control fat phase. Among all the five types of fats and oils tested, pork lard contains the highest level of PUFA, followed by Menhaden fish oil, canola oil, beef tallow, and chicken fat. The immediate reduction in *Salmonella* to a non-detectable level (0 h) from both aqueous and fat phases of Menhaden oil and lard corresponds to their high level of PUFA. The antimicrobial effect of arachidonic acid (C-20) exhibited a direct relation with the increasing level of unsaturation, with C-20:5 showing the highest reduction and C-20:1 showing the lowest reduction against *S.* aureus [30] when incubated at room temperature for 1 h. The authors also discovered a similar relation between the reduction in *S*. aureus and the degree of unsaturation with Oleic acid (C-18). Docosahexaenoic acid (DHA), a PUFA from *Sardinell longiceps* and *Sardinells fimbriata*, showed potent antibacterial activity against *Salmonella* and *E. coli* [34] in an in vitro assay at 37 °C. A similar efficacy of Menhaden oil and lard was also observed in the case of untreated (control) samples, where the quickest reduction in *Salmonella* to a non-detectable level was reported as compared to chicken fat, canola oil, and beef tallow (Table 3).

Lamb [32] reported that a cocktail of *Salmonella* (*S*. Typhimurium (ATCC 13312), *S*. choleraesuis subsp. choleraesuis (ATCC 13311), *S*. Pullorum (ATCC 19945), and *S*. Choleraesuis subsp. arizonae (ATCC 13314)) survived the 7-day incubation period at 26 °C in duck fat, beef tallow, and pig lard. This finding was contrary to our finding where control fat phase (*Salmonella*-inoculated fats without antimicrobial addition) reduced the *Salmonella* cocktail to a non-detectable level in 2 h. The difference in the results could be attributed to the method and enrichment techniques. Our experimental setup simulated the real-life bulk fat storage and transport system by providing ~3% moisture in the bottom aqueous layer and a top fat layer. The author did not incorporate the aqueous layer. Secondly, we inoculated *Salmonella* as wet inoculum, which could have led to the sedimentation of the pathogen to the aqueous phase. Finally, the enrichment of the inoculated fats before plating could have caused the revival of the stressed cells. In our previous study in chicken fat, we discovered similar findings wherein the fat phase, which was negative for *Salmonella* on the agar plate, was positive for *Salmonella* upon enrichment followed by PCR confirmation [22]. In another study by Molitor et al. [25], the authors reported that eight logs CFU/mL of *Salmonella* cocktail (*S*. Senftenberg, *S*. Newport, *S*. Thompson, and *S*. Infantis) was reduced to a non-detectable level by day 3 at 48 °C in both choice white grease and beef tallow. The longer survival time in this study could partly be the effect of sampling techniques, where the authors took mixed samples from both the fat phase and water phase. In our study, the separate sampling of the two phases in the fat system yielded a non-detectable *Salmonella* in 2 h in the fat phase and consistently higher (approximately eight logs) counts throughout the experimental period in the aqueous phase. Therefore, an enrichment followed by a molecular confirmation is necessary to call a fat and oil sample negative [22]. 

## 5. Conclusions

In summary, the findings from this study suggest that coating pet foods with rendered animal fats and oils could help in mitigating post-processing *Salmonella* contamination. The addition of acidulants in the fat and oil system boosts the antimicrobial efficacy of fats and oils. A future study involving the palatability test of these fats and oil-coated dog food is warranted.

## Figures and Tables

**Figure 1 animals-13-01304-f001:**
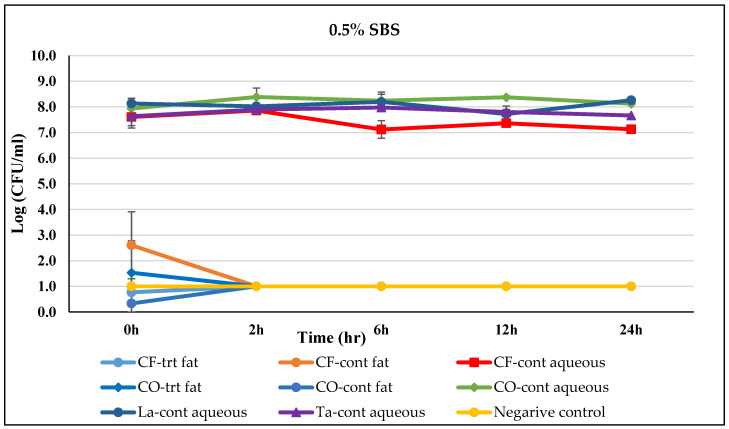
Mean logarithmic counts (log CFU/mL) of *Salmonella* spp. in different fat and oil systems with and without the inclusion of 0.5% sodium bisulfate evaluated separately for the aqueous phase and fat phase at 45 °C. Treatments from each phase were plated on TSA at different times. SBS, sodium bisulfate; CF, chicken fat; CO, canola oil; La, Lard; Ta, Tallow; Cont., control. Negative control consisted of fat-oil system without an acidulant and *Salmonella* inoculation. The limit of detection is 1 log CFU/mL for this study. Because the three replications were averaged, for some of the treatments, the counts on 0 h appeared to be lower-than-detection limit. Error bars are ±1 standard error of the mean.

**Figure 2 animals-13-01304-f002:**
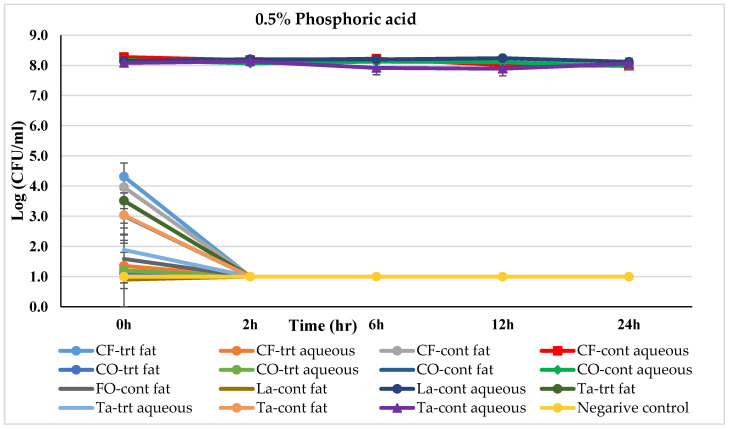
Mean logarithmic counts (log CFU/mL) of *Salmonella* spp. in different fat and oil systems with and without the inclusion of 0.5% phosphoric acid evaluated separately for the aqueous phase and fat phase at 45 °C. Treatments from each phase were plated on TSA at different times. SBS, sodium bisulfate; CF, chicken fat; CO, canola oil; FO, fish oil; La, Lard; Ta, Tallow; Cont., control. Negative control consisted of fat-oil system without acidulant and *Salmonella* inoculation. The limit of detection is 1 log CFU/mL for this study. Because the three replications were averaged, for some of the treatments, the counts on 0 h appeared to be lower-than-detection limit. Error bars are ±1 standard error of the mean.

**Figure 3 animals-13-01304-f003:**
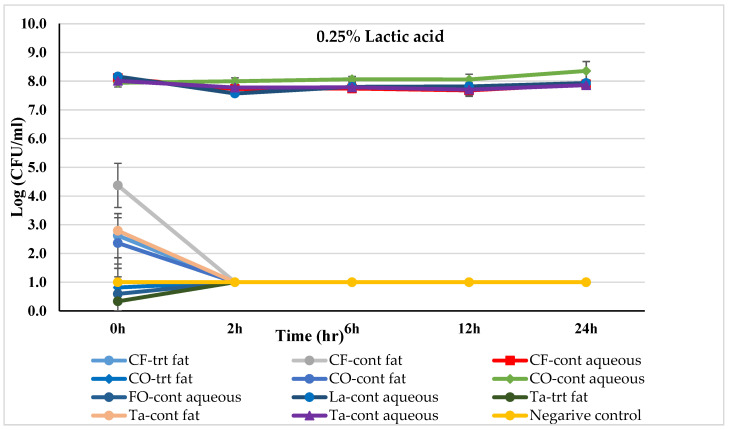
Mean logarithmic counts (log CFU/mL) of *Salmonella* spp. in different fat and oil systems with and without the inclusion of 0.25% lactic acid, evaluated separately for the aqueous phase and fat phase at 45 °C. Treatments from each phase were plated on TSA at different times. SBS, sodium bisulfate; CF, chicken fat; CO, canola oil; FO, fish oil; La, Lard; Ta, Tallow; Cont., control. Negative control consisted of fat-oil system without acidulant and *Salmonella* inoculation. The limit of detection is 1 log CFU/mL for this study. Because the three replications were averaged, for some of the treatments, the counts on 0 h appeared to be lower-than-detection limit. Error bars are ±1 standard error of the mean.

**Figure 4 animals-13-01304-f004:**
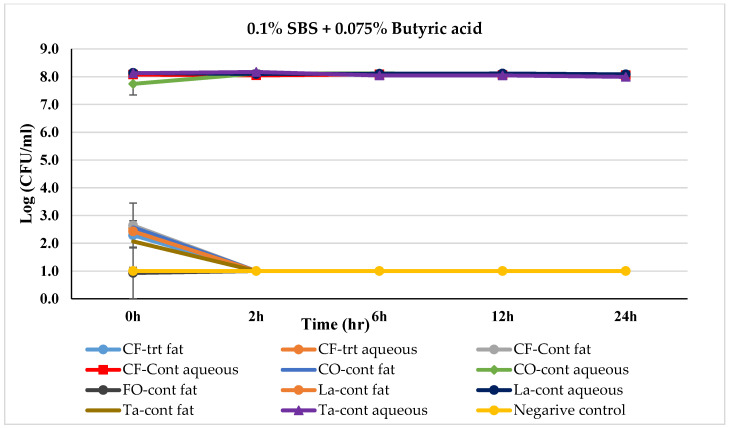
Mean logarithmic counts (log CFU/mL) of *Salmonella* spp. in different fat and oil systems with and without the inclusion of 0.1% SBS and 0.075% butyric acid evaluated separately for the aqueous phase and fat phase at 45 °C. Treatments from each phase were plated on TSA at different times. SBS, sodium bisulfate; CF, chicken fat; CO, canola oil; FO, fish oil; La, Lard; Ta, Tallow; Cont., control. Negative control consisted of fat-oil system without acidulant and *Salmonella* inoculation. The limit of detection is 1 log CFU/mL for this study. Because the three replications were averaged, for some of the treatments, the counts on 0 h appeared to be lower-than-detection limit. Error bars are ±1 standard error of the mean.

**Figure 5 animals-13-01304-f005:**
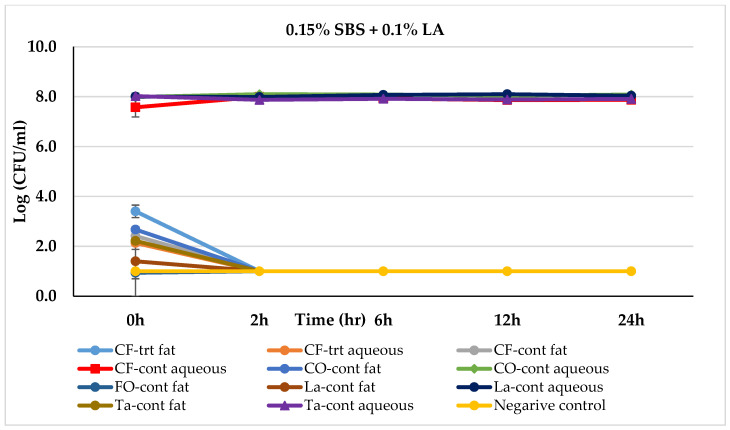
Mean logarithmic counts (log CFU/mL) of *Salmonella* spp. in different fat and oil systems with and without the inclusion of 0.15% SBS and 0.1% lactic acid evaluated separately for the aqueous phase and fat phase at 45 °C. Treatments from each phase were plated on TSA at different times. SBS, sodium bisulfate; CF, chicken fat; CO, canola oil; FO, fish oil; La, Lard; Ta, Tallow; Cont., control. Negative control consisted of fat-oil system without acidulant and *Salmonella* inoculation. The limit of detection is 1 log CFU/mL for this study. Because the three replications were averaged, for some of the treatments, the counts on 0 h appeared to be lower-than-detection limit. Error bars are ±1 standard error of the mean.

**Figure 6 animals-13-01304-f006:**
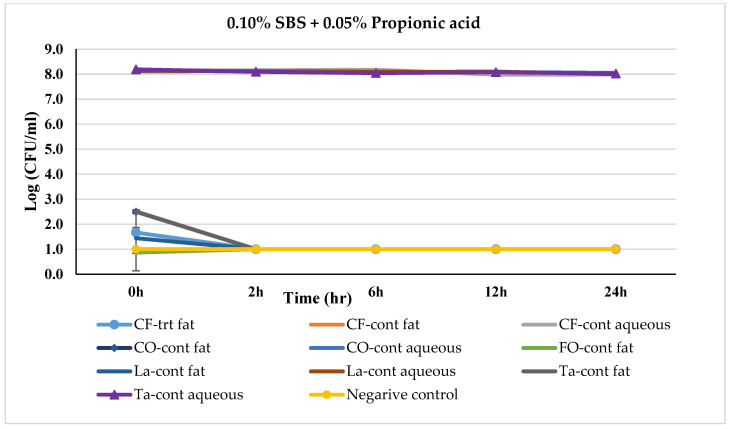
Mean logarithmic counts (log CFU/mL) of *Salmonella* spp. in different fat and oil systems with and without the inclusion of 0.10% SBS and 0.05% propionic acid evaluated separately for the aqueous phase and fat phase at 45 °C. Treatments from each phase were plated on TSA at different times. SBS, sodium bisulfate; CF, chicken fat; CO, canola oil; FO, fish oil; La, Lard; Ta, Tallow; Cont., control. Negative control consisted of fat-oil system without acidulant and *Salmonella* inoculation. The limit of detection is 1 log CFU/mL for this study. Because the three replications were averaged, for some of the treatments, the counts on 0 h appeared to be lower-than-detection limit. Error bars are ±1 standard error of the mean.

**Table 1 animals-13-01304-t001:** Time to non-detection of *Salmonella* in acidulants-treated fat or oil system.

Acidulant Treatments	Chicken Fat	Canola Oil	Fish Oil	Lard	Tallow
0.50% sodium bisulfate (SBS)	2 h/0 h *	2 h/0 h	0 h/0 h	0 h/0 h	0 h/0 h
0.50% phosphoric acid	2 h/2 h	2 h/2 h	0 h/0 h	0 h/0 h	2 h/2 h
0.25% lactic acid	2 h/0 h	2 h/0 h	0 h/0 h	0 h/0 h	2 h/0 h
0.10% SBS + 0.075% butyric acid	2 h/2 h	0 h/0 h	0 h/0 h	0 h/0 h	0 h/0 h
0.15% SBS + 0.10% lactic acid	2 h/2 h	0 h/0 h	0 h/0 h	0 h/0 h	0 h/0 h
0.10% SBS + 0.05% propionic acid	2 h/0 h	0 h/0 h	0 h/0 h	0 h/0 h	0 h/0 h

* X/Y = fat phase/aqueous phase.

**Table 2 animals-13-01304-t002:** Minimum Inhibitory Concentrations (MIC) of acidulants against *Salmonella* spp.

Acidulant Antimicrobials	Serotypes	MIC	MIC for Cocktail Serotypes *
Sodium bisulfate (SBS)	*S*. Enteritidis (ATCC 4931)	0.31%	0.31%
	*S*. Heidelberg (ATCC 8326)	0.31%	
	*S*. Typhimurium (ATCC 14028)	0.50%	
Lactic acid (LA)	*S*. Enteritidis (ATCC 4931)	0.20%	0.20%
	*S*. Heidelberg (ATCC 8326)	0.20%	
	*S*. Typhimurium (ATCC 14028)	0.50%	
Phosphoric acid (PA)	*S*. Enteritidis (ATCC 4931)	0.10%	0.20%
	*S*. Heidelberg (ATCC 8326)	0.10%	
	*S*. Typhimurium (ATCC 14028)	0.25%	
SBS + butyric acid	*S*. Enteritidis (ATCC 4931)	0.10% + 0.05%	0.1 + 0.08%
	*S*. Heidelberg (ATCC 8326)	0.10% + 0.08%	0.05% + 0.10%
	*S*. Typhimurium (ATCC 14028)	0.10% + 0.05%	
SBS + lactic acid	*S*. Enteritidis (ATCC 4931)	0.10% + 0.15%,	0.15% + 0.10%
		0.05% + 0.10%	
	*S*. Heidelberg (ATCC 8326)	0.15% + 0.10%	
		0.10% + 0.10%	
		0.05% + 0.15%	
	*S*. Typhimurium (ATCC 14028)	0.10% + 0.10%	
		0.05% + 0.15%	
SBS + Propionic acid	*S*. Enteritidis (ATCC 4931)	0.10% + 0.05%	0.10% + 0.05%
		0.05% + 0.10%	0.05% + 0.10%
	*S*. Heidelberg (ATCC 8326)	0.10% + 0.05%	
		0.05% + 0.10%	
	*S*. Typhimurium (ATCC 14028)	0.10% + 0.05%	
		0.05% + 0.08%	

* cocktail = *S.* Typhimurium, *S.* Heidelberg, and *S.* Enteritidis.

**Table 3 animals-13-01304-t003:** Time to non-detection of *Salmonella* in control (without acidulant) fat or oil system.

	Chicken Fat	Canola Oil	Fish Oil	Lard	Tallow
Fat phase	1.67 h *	1.67 h	0.67 h	1 h	1.67 h
Water phase	>24 h	>24 h	0.25 h	>24 h	>24 h

* Time was calculated averaging the controls (fat system without acidulant treatment) from 6 sets of experiments.

## Data Availability

Data will be provided upon reasonable request.

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
