# Peer review of "Application of Acidulants to Control Salmonella spp. in Rendered Animal Fats and Oils with Different Levels of Unsaturation"

_animals, 2023, doi:10.3390/ani13081304_

Round 1

Reviewer 1 Report

The research talks about controlling Salmonella in pet food with the aid of acidulants. Reviewer appreciates the author's contribution to the rendering and pet food industry by addressing the niche environment which is more likely to harbor pathogenic microorganisms.

Here are some suggestions-

Ln 86 – Define SBS

Ln 128 – What is water phase fat/oil type?

Ln 147 – For the factorial arrangement, how about the 6 treatment types?

For LOD, what scientific or statistical bases were used to determine the 1 Log minimum concentration? Will this value be enough to justify a zero-tolerance policy in the pet food industry?

In the figures, does the purple line represent an initial inoculum concentration? It must be expressed somewhere in the figure if that is the case. Same with LOD – Yellow line in the graphs.

Author Response

  1. Ln 86 – Define SBS

Response: The recommended edit is made. Thank you.

  1. Ln 128 – What is water phase fat/oil type?

Response: The information is clarified and incorporated into the text. Thank you.

  1. Ln 147 – For the factorial arrangement, how about the 6 treatment types?

Response: We agree with your suggestion. Including a negative control, it is a 6 x 5 factorial arrangement. The correction is made. Thank you.

  1. For LOD, what scientific or statistical bases were used to determine the 1 Log minimum concentration? Will this value be enough to justify a zero-tolerance policy in the pet food industry?

Response: The limit of detection (LOD) was calculated based on the volume of the suspension plated and the number of detectable colonies. For example, while plating 100 µL onto an agar plate, 1 is the minimum number of detectable colonies. Similarly, for 1000 µL (or 1 mL or per mL) volume, the detectable colonies would be 10, which is equivalent to 1 log. Therefore, the LOD was set as 1 log.

When it comes to the zero-tolerance policy in pet foods, it means the absence of any detectable pathogen from the test sample. A result below the LOD would be expressed as <1 log CFU/ml. I hope this helps to satisfy your concern. We personally think this piece of comment might not be too relevant to incorporate into the main text. Therefore, it is not included. Thank you.

Reviewer 2 Report

Dear Authors,

Thank you for submitting this manuscript that explores the use of acidulants to control Salmonella prevalence in rendered animal fats and oils. This is an interesting study with a lot of potential application.

There are some small revisions required pertaining to grammar and formatting. I have added these comments to the PDF version of the paper.

Author Response

Response: Thank you for your suggestions. The comments and suggestions provided in the PDF file are addressed and incorporated in the revised submission. A track-change copy of the edited manuscript is also attached along with the clean revised copy.

Reviewer 3 Report

Interesting topic, but I have microbiological doubts related to the reduction of the Salmonella number after 0 hours from introducing them to the water and fat phase of rendered fat. If the authors considered the microbial growth curve, they should have known that if microbes change their environment after inoculation, they must adapt to the new one. This was probably the reason for not detecting them after inoculation, when, for example, after 2 hours some contamination was detected. This is a general note to the text.
Here are other comments and requests for clarification:

1. Introduction: I would like the Authors to consistently use the term "acidulants" without "chemicals". This last word is perceived negatively, regardless of whether we are talking about food for humans or animals.

2. Materials and Methods: lines 122-123 - was the MIC assessed by turbidity only? This is sometimes very difficult and reagents such as an aqueous solution of 2, 3, 5-triphenyltetrazolium chloride (INT) are used for this purpose. Then there is no doubt, because the change of color is very clear.

3. Who is the author of the method described in "2.3. Survival of Salmonella in fats and oils"? What is the reason for the temperature of 45 degrees Celsius, in which fats and acidulants were incubated overnight? Please complete this information.

4. How were Salmonella cocktails prepared? Please provide more details (culture volumes, cell density, etc.).

5. Since the experiment was repeated three times, why is there no statistical analysis of the results? It is difficult to do with MIC, but wherever the number of bacteria is marked (or the log of the number), it should be statistically developed.

6. Figures: I have some confusion about figures, because they are not very legible (the lines overlap) and additionally they differ in the number of lines. Thus, there are more lines in Fig. 1 than variants signed in the description (there are 6 variants). There are 9 variants in figures 2, 3 and 4. Again, figure 6 has a caption with 11 variants. Please check if this is the correct record and make any corrections.

7. I consider tables 2 and 3 redundant, because in my opinion these results are contained in the graphs. Unless I'm wrong? And the captions are nonclear: "non-detection" but of what?

8. References: please check the records carefully, because some are written differently than others (e.g. numbers: 5, 24, 25). Sometimes full journal names are given and article pages are missing. The dots at the end of the record are either there or not. The record should comply with the requirements of the Editorial Office.

Author Response

  1. Interesting topic, but I have microbiological doubts related to the reduction of the Salmonella number after 0 hours from introducing them to the water and fat phase of rendered fat. If the authors considered the microbial growth curve, they should have known that if microbes change their environment after inoculation, they must adapt to the new one. This was probably the reason for not detecting them after inoculation, when, for example, after 2 hours some contamination was detected. This is a general note to the text.

Response: Thank you for the comment. We agree with your point that bacteria need adaptation to a new environment. Because the initial inoculum was already ~ 8 logs,

It is very unlikely for all of them to go undetectable as quickly as 0 h or 2 h just because they are adapting to a new environment. Rather, in our study, for many of the treatments they were actually detectable at 0 h, as shown in figures 1 through 6. I hope this clarifies. Thank you.

  1. Introduction: I would like the Authors to consistently use the term "acidulants" without "chemicals". This last word is perceived negatively, regardless of whether we are talking about food for humans or animals.

Response: Thank you for the suggestion. This makes sense! The necessary corrections are made.

  1. Materials and Methods: lines 122-123 - was the MIC assessed by turbidity only? This is sometimes very difficult and reagents such as an aqueous solution of 2, 3, 5-triphenyltetrazolium chloride (INT) are used for this purpose. Then there is no doubt because the change of color is very clear.

Response: We agree with your comment that visual turbidity is sometimes difficult especially when we have an opaque chemical, such as medium chain fatty acids or liquid smoke. However, in this case, as the acidulants were clear solutions, we used the conventional use visual method. Thank you.

  1. Who is the author of the method described in "2.3. Survival of Salmonella in fats and oils"? What is the reason for the temperature of 45 degrees Celsius, in which fats and acidulants were incubated overnight? Please complete this information.

Response: The same authors are the authors for method 2.3. The reason the fats or oil systems were incubated at 45° C is that at that temperature all the fats were in a molten state. Because we have two phases that needed to be analyzed separately, we need a molten fat or oil system. This information is incorporated in the manuscript to make it clearer. Thank you.

  1. How were Salmonella cocktails prepared? Please provide more details (culture volumes, cell density, etc.).

Response: Thank you for your question. This was prepared on a volume basis. The information is incorporated in the materials and methods section. Thank you.

  1. Since the experiment was repeated three times, why is there no statistical analysis of the results? It is difficult to do with MIC, but wherever the number of bacteria is marked (or the log of the number), it should be statistically developed.

Response: The three replicates were used to take the average, plot the table and graphs, and create error bars.

All the treatments (plus the fish oil controls) were effective to lower the Salmonella count to a non-detectable limit by 2 hours. This time duration is very short and extremely effective because it is practically impossible for the treated fats and oils can reach to the end consumer (dogs) or handler (human) before 2 hours. Therefore, statistical analysis on the reduction comparisons for just one-time points (0 h) is not of practical significance. Thank you.

  1. Figures: I have some confusion about the figures because they are not very legible (the lines overlap) and additionally they differ in the number of lines. Thus, there are more lines in Fig. 1 than variants signed in the description (there are 6 variants). There are 9 variants in Figures 2, 3, and 4. Again, figure 6 has a caption with 11 variants. Please check if this is the correct record and make any corrections.

Response: Thank you for putting this question. It is mentioned in the second sentence on 3.2 materials and methods that only the treatments with detectable Salmonella past 0 h are shown in graphs. Therefore, for some treatments (means in some figures), some specific fat types treatments are non-detectable immediately (0 h) and those are not represented in graphs. Therefore, the number of line representations is not uniform across the figures. Thank you.

  1. I consider tables 2 and 3 redundant because in my opinion these results are contained in the graphs. Unless I'm wrong? And the captions are nonclear: "non-detection" but of what?

Response: Due to the nature of the graphs with multiple treatments and controls, tabulating just the ‘non-detection’ time for Salmonella in acidulant treated and controls would become easy for readers to catch.

‘Non-detection’ is for Salmonella. This information is incorporated in table captions, to make it clearer. Thank you.

  1. References: please check the records carefully, because some are written differently than others (e.g., numbers: 5, 24, 25). Sometimes full journal names are given and article pages are missing. The dots at the end of the record are either there or not. The record should comply with the requirements of the Editorial Office.

Response: Thank you for pointing that out. Necessary corrections are made according to the journal requirement.
